# Low bone mass in people living with HIV on long-term anti-retroviral therapy: A single center study in Uganda

Erisa Sabakaki Mwaka[1,2]*, Ian Guyton Munabi[1], Barbara Castelnuovo[2], Arvind Kaimal[2], William Kasozi[2], Andrew Kambugu[1], Philippa Musoke[3], Elly Katabira[4]

1 School of Biomedical Sciences, College of Health Sciences, Makerere University, Kampala, Uganda,
2 Infectious Diseases Institute, College of Health Sciences, Makerere University, Kampala, Uganda,
3 Department of Paediatrics and Child Health, School of Medicine, College of Health Sciences, Makerere University, Kampala, Uganda, 4 Department of Medicine, School of Medicine, College of Health Sciences, Makerere University, Kampala, Uganda

* erisamwaka@gmail.com

## Abstract

**Data Availability Statement:** All data files are available from the Figshare database (DOI:10.6084/m9.figshare.13174628).

### Background

This study set out to determine the prevalence of low bone mass following long-term exposure to antiretroviral therapy in Ugandan people living with HIV.

### Methods

A cross-sectional study was conducted among 199 people living with HIV that had been on anti-retroviral therapy for at least 10 years. All participants had dual X-ray absorptiometry to determine their bone mineral density. The data collected included antiretroviral drug history and behavioral risk data Descriptive statistics were used to summarize the data. Inferential statistics were analyzed using multilevel binomial longitudinal Markov chain Monte Carlo mixed multivariate regression modelling using the *rstanarm* package.

### Results

One hundred ninety nine adults were enrolled with equal representation of males and females. The mean age was 39.5 (SD 8.5) years. Mean durations on anti-retroviral treatment was 12.1 (SD 1.44) years, CD4 cell count was 563.9 cells/mm$^3$. 178 (89.5%) had viral suppression with <50 viral copies/ml. There were 4 (2.0%) and 36 (18%) participants with low bone mass of the hip and lumbar spine respectively. Each unit increase in body mass index was associated with a significant reduction in the odds for low bone mineral density of the hip and lumbar spine. The duration on and exposure to the various antiretroviral medications had no significant effect on the participant's odds for developing low bone mass. All the coefficients of the variables in a multivariable model for either hip or lumbar spine bone mass were not significant.

**Funding:** This work was supported by University of Alabama at Birmingham (UAB) Center For AIDS Research in the form of a grant awarded to ESM (000503356-SP002-24), which was a sub-award from a National Institutes of Health grant awarded to Principal Investigator Michael Saags (P30AI027767-28). Janssen pharmaceutical company provided support through a Johnson & Johnson corporate citizenship trust used to partly fund the ALT cohort at the Infectious Disease Institute-Uganda. The funders had no role in study design, data collection and analysis, decision to publish, or preparation of the manuscript.

**Competing interests:** We declare competing interest; Janssen pharmaceutical company provides financial support to the Antiretroviral Treament Longterm (ALT) Cohort at the Infectious Diseases Institute-Uganda. However, this does not alter our adherence to PLOS ONE policies on sharing data and materials.

## Conclusion

These results provide additional evidence that patients on long term ART achieve bone mass stabilization. Maintaining adequate body weight is important in maintaining good bone health in people on antiretroviral therapy.

## Introduction

Globally, nearly 70% of the people living with the human immunodeficiency virus (HIV), which is an estimated 37.9 million people, are in Sub-Saharan Africa (SSA) [1]. As of June 2019, 24.5 million people were accessing antiretroviral therapy and currently the World Health Organization (WHO) recommends that all HIV infected individuals should be initiated on ART upon diagnosis [2]. The wide spread use of antiretroviral therapy (ART) to treat HIV infection has led to increased lifespans of HIV-infected persons and a marked reduction in AIDS associated complications but created additional health challenges related to aging with HIV [3,4].

The burden of non-communicable diseases (NCDs) in sub-Saharan Africa (SSA) is higher than the global average; and should therefore be prioritized on the health and development agenda [5]. Non-communicable diseases are becoming a leading cause of morbidity and mortality also in the HIV-infected populations especially those on long-term anti-retroviral therapy (ART) [6]. Long-term use of ART has been associated with several adverse effects on bone mineral density (BMD) that call for continual monitoring, especially now that a sizable proportion of Ugandan people living with HIV (PLWH) have been on ART for long durations. It is important to note that 67% of adults living with HIV are accessing ART in eastern and southern Africa [1] with a sizable proportion of these being on ART for over a decade [6], yet relatively little is known about the impact of ART on bone health.

The prevalence of low BMD in PLWH and ART-treated individuals is reported to be twice that in health individuals [7]. The prevalence of osteopenia and osteoporosis in PLWH ranges from 48% to 55% and 10% to 34% respectively [8,9] with HIV-infected individuals having a 6.3 times greater chance of developing low BMD when compared to the HIV-uninfected, and HIV infected individuals on ART-treated having a 2.5 times greater risk of developing low BMD compared to the ART-naïve individuals [3]. Results form a meta-analysis [7] showed that HIV infected individuals had twice the risk of developing hip and lumbar spine low BMD (odds ratio 2.4 and 2.6 respectively), compared to the HIV-uninfected. In the same meta-analysis, the odds of developing low BMD at the hip and lumbar spine was 2.8 times (p = 0.004) and 3.4 times (p = 0.0002) respectively when compared to ART-naïve individuals.

Generally, BMD is higher among PLWH in low and middle-income countries (LMIC) compared to resource rich settings with prevalence rates as high as 85% [10–14]. Data from developed countries consistently show that HIV infection is associated with low BMD and an increased fracture risk, however, there is limited published data from resource limited settings (RLS) on longitudinal changes in BMD, the effect of ART regimens on BMD and almost no data on fragility fractures among HIV infected individuals. This could be attributed to bone health not being a priority in these settings, limited availability of dual energy X-ray absorptiometry (DXA) machines and the prohibitive cost of DXA scanning in most SSA countries [14].

The risk factors for low BMD in the developed world remain similar to those in RLS [8,15,16]. However, some of these risk factors such as low body mass index (BMI),

malnutrition, advanced disease, longer duration of disease and a higher viral load are more prevalent in RLS [10,11,17,18]. The prevalence of low BMD in the Ugandan population is not known. Furthermore, there are no local guidelines for the diagnosis and management of osteoporosis in the country. This study therefore set out to determine the prevalence and predictors of low BMD following long-term exposure to ART in Ugandan PLWH.

Much as many studies with relatively young populations [8,9,17,19] use the World Health Organization criteria for categorizing bone loss, in this study we used the International Society of Clinical Densitometry (ISCD) official position that recommends the use of "Bone mass" instead of "Bone mineral density" for pre-menopausal females, males younger than 50 years and children [20].

## Materials and methods

### Study design and setting

This was a cross-sectional study conducted over an 18 month period in a cohort of PLWH receiving treatment from the Makerere University Infectious Diseases Institutes (IDI) HIV care clinics in Kampala, the capital city of Uganda. The IDI is a Center of Excellence for HIVAIDS research and service delivery in Uganda [21]. The clinic opens 10 hours a day, 5 days a week and currently supports the care for over 8,000 HIV/AIDS clients.

The study population comprised of adult PLWH, who are part of the IDI Antiretroviral Treatment Long-term (ALT) cohort, which has been enrolling patients receiving ART consecutively for at least 10 years and plan to follow them up for another 10 years. The ALT cohort currently comprises 1000 individuals who have been undergoing annual evaluations for more than 10 years. Details of the ALT cohort have been published and described elsewhere [22]. Using randomly generated computer numbers we randomly enrolled adults above 18 years who had been receiving ART for at least 10 years and were willing and able to comply with all study procedures. We excluded clients who were hospitalized or bedridden, enrolled in clinical trials, those with pathological fractures, history of trauma or surgery at the measurement site and those that were unwilling or unable to comply with any part of the study protocol. One hundred ninety nine PLWH were enrolled in the study (Fig 1).

The sample size was calculated using the www.openepi.com online proportions sample size calculator for: an 80.4% prevalence of low BMD among HIV positive persons on ART [11], population size of 1000 HIV positive clients on ART, confidence limits of 5% and design effect of 1.3 arising from the multiple medications for each client at any one time to give a calculated sample size of 190 respondents for a power of 90. To this was included a 5% allowance for loss and omissions, binging the final sample size to 200 respondents.

Participants were consecutively recruited over the 18-month study period. Clients were invited to participate and enrolled during their routine annual clinic visit.

### Study procedures

All participants were subjected to lumbar spine, left total hip and left femoral neck body scans, to assess body mineral content and bone mineral density (BMD), using dual energy X-ray absorptiometry (DXA) (Hologic Discovery Wi Apex 3.1, Hologic Bedford Inc., Bedford, MA, USA) using standard protocols. Bone mineral density was expressed in grams of mineral per square centimetre. The reference population for this scanning was age, sex, race and BMI matched from the National Health and Nutrition Examination Survey (NHANES) cohort [23]. Bone mass was categorized using the official position of the International Society of Clinical Densitometry (ISCD) since most participants were below 50 years of age [20]. The ISCD official position recommends the use of the Z-score for pre-menopausal females, males

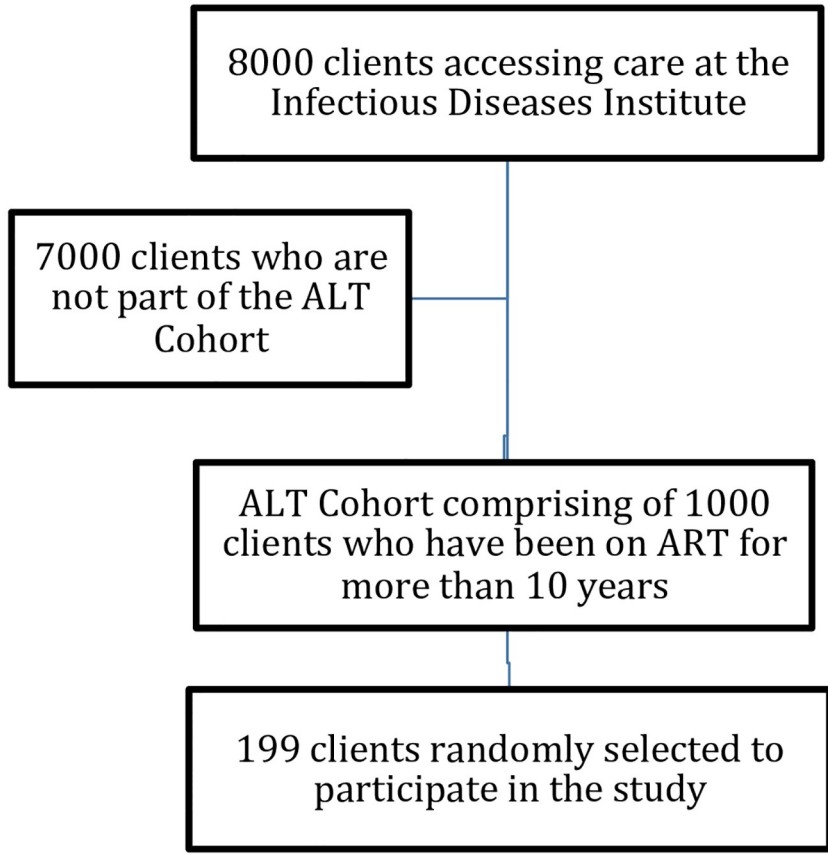

**Fig 1. Participant recruitment.**

younger than 50 years and children. A Z-score of -2.0 or lower was defined as having low bone mass for age and a Z-score above -2.0 was defined as bone mass within the expected range for age [20]. The World Health Organization (WHO) was used for post-menopausal females and males above 50 years of age [24,25]. A T-score that was within one standard deviation (SD) of the reference BMD was classified as normal, 1 to 2.5 SD below the reference BMD as osteopenia, and greater than 2.5 SD below the reference BMD as osteoporosis. For this study all participants with low bone mass or osteopenia/osteoporosis were categorized as having "low bone mass" while the rest were categorized as having normal bone mass (BM). This was done for ease of data analysis. The following data was also collected at enrolment: age, sex, height and weight for use in calculating Body Mass Index (BMI), the type of ART (categorized as Nucleoside reverse transcriptase inhibitors, Non-nucleoside reverse transcriptase inhibitors, protease inhibitors, and Integrase inhibitors), duration on specific antiretroviral agents, behavioural risk information including alcohol and tobacco use, and history of previous fractures of the wrist, hip or spine, using a study questionnaire and data abstraction tool.

### Data collection and statistical analysis

Data were collected initially collected using case report forms and then entered into DataFax forms that were specially designed for this study. This data management system is designated to manage paper data forms. The forms are faxed to the DataFax server where they are read

using intelligent characters-recognition and then added to the study database. The latest viral load results and nadir CD4 count were retrieved from the Integrated Clinic Enterprise Application (ICEA) database, an in-house software that provides good quality data collection, with minimal missing or incorrect information [26]. These validated records were then exported as excel files for transformation so that each row represented an individual drug exposure prior to further analysis in the R-statistical computing environment [27]. The final long dataset comprised of 1384 anti-retroviral drug exposures. The analysis was informed by the study hypothesis that: *exposure (time on treatment and type of drugs) to ART was associated with differences in BM among respondents after controlling for age, alcohol consumption, sex, smoking, most recent viral load, CD4 cell count, line of treatment (switch) and body mass index.* During analysis descriptive statistics were generated and summarised as frequencies and correlations in tables. Inferential statistics were generated with multilevel binomial longitudinal Markov chain Monte Carlo (MCMC) mixed multivariate regression modelling using the *rstanarm* package [28–30]. The diagnostics for each model, one for the spine and other for hip BM, were used to identify any poorly performing values and assess the sampling quality. The absence of a value of "zero" in the 95% confidence interval for the coefficients was used to identify statistically significant output values. The mean coefficients were summarised in tables but exponentiated to obtain odds ratios. All records with missing data were removed from the database prior to the inferential MCMC part of the analysis. Ethics approval was obtained from the Makerere University School of Biomedical Sciences Higher Degrees and Research Ethics Committee (SBSHDREC 410) and the Uganda National Council of Science and Technology (HS 2179). All participants provided informed consent to participate in the study.

## Results

One hundred ninety nine PLWH were recruited between July 2017 and July 2018. Fewer participants (4/199, 2%) had left hip low bone mass compared to the lumbar spine (36/199, 18%) as shown in Table 1. There was equal representation of both males (50%) and females (50%). The mean age was 39.5 (SD 8.5) years and female participants were on average younger than male participants by 1.38 years (standard error = 1.21, t-value = -1.14, P-value = 0.25). At the time of recruitment, participants had on average been living with HIV for 12.6 (SD 1.55) years since diagnosis and a mean of 12.1 (SD 1.44) years on ART. One hundred seventy eight (89.5%) of the participants had viral suppression with <50 viral copies/ml. The mean CD4 cell count was 564 cells/mm$^3$ (SD 20.4, CI 524–604).

The correlations between the study variables as summarized in Table 2, were moderate to weak. There was moderate correlation for observed low BM between the hip and the lumbar spine (R = 0.30, P-value = <0.01).

Fig 2 provides a summary of the different ART drug classes in the various prescriptions for each participant, from the time of initiation on ART till recruitment into the study. From this table non-reverse transcriptase inhibitors (NRTI) were the most commonly prescribed medications.

There were three incomplete participants records 4/199 (2%), which were excluded, leaving 196 complete respondents' records for further analysis.

Table 3 provides a summary of the best-fitted univariable and multivariable MCMC models for BMI determined using hip BM and lumbar spine BM. Overall on the univariable analysis, each unit change in the classification of BMI was associated with a large reduction in the odds for low BM. This was significant for both the hip (OR = 0.06, 95% CI 0.01 to 0.47) and lumbar spine (OR = 0.004, 95% CI 0.0002 to 0.07). Also, in this table note that, for both hip (OR = 70.11, 95% CI 1.75 to 3,071) and lumbar spine (OR = 14,185, 95% CI 20.70 to

**Table 1. Descriptive statistics for the study population.**

| Variable | Observation | Total (%) |
|---|---|---|
| Age | Mean (SD) | 39.4 (8.5) |
| Female gender | Yes | 100 (50.3) |
| | No | 99 (49.7) |
| Alcohol consumption | Yes | 37 (18.6) |
| | No | 162 (81.4) |
| Smoke | No | 177 (88.9) |
| | Yes | 22 (11.1) |
| Left Hip BM | Low BMD | 4 (2.0) |
| | Normal | 195 (98.0) |
| Lumbar Spine BM | Low BMD | 36 (18.1) |
| | Normal | 163 (81.9) |
| Body Mass Index* | Normal weight | 108 (55.1) |
| | Overweight | 57 (29.1) |
| | Obese | 16 (8.2) |
| | Under weight | 15 (7.7) |
| Viral load | Mean (SD) | 19568.8 (77277.2) |
| | Suppressed | 178 (89.45) |
| | Not-Suppressed | 21 (10.55) |

*The WHO general cut off points for BMI classification were used for this study however literature suggests that they may vary with race/ethnicity and between countries [31–33].

18,255,921), participants who were underweight were several times more likely to have a low BM compared to the normal weight participants. On the other hand, participants who were overweight were less likely to have low BM. This was significant for the lumbar spine observations (OR = 0.002, 95% CI <0.01 to 0.29). Also, either being female or each unit increase in age was associated with significantly lower odds of low BM on univariable modeling. All the coefficients for the variables of the best multivariable model for both hip and lumbar spine BM were not significant. As shown in Table 3, for both the univariable and multivariable modeling, the duration on and exposure to the various ART medications had no significant effect on the participant's odds for developing low BM. For the multivariable model interactions in Table 3,

**Table 2. Showing the correlations between the study variables.**

| | Age | Gender | Alcohol | Smoking | Lumbar BM | Left Hip BM | Body mass index | Viral suppression |
|---|---|---|---|---|---|---|---|---|
| Age | | | | | | | | |
| Gender | -0.08 | | | | | | | |
| Alcohol | 0.01 | -0.01 | | | | | | |
| Smoking | 0.16* | 0.13 | 0.24*** | | | | | |
| Lumbar BM | 0.19** | 0.21** | -0.09 | -0.08 | | | | |
| Left hip BM | -0.05 | 0.00 | -0.07 | 0.05 | 0.30**** | | | |
| Body mass index | 0.09 | 0.13 | 0.05 | -0.07 | 0.02 | -0.07 | | |
| Viral suppression | -0.02 | 0.02 | -0.04 | -0.02 | -0.03 | -0.05 | -0.06 | |
| CD4 | 0.08 | 0.13 | -0.03 | -0.05 | 0.11 | 0.05 | 0.10 | 0.25** |

Stars Key: $p < .0001$, "****", $p < .001$, "***", $p < .01$, "**", $p < .05$, "*".

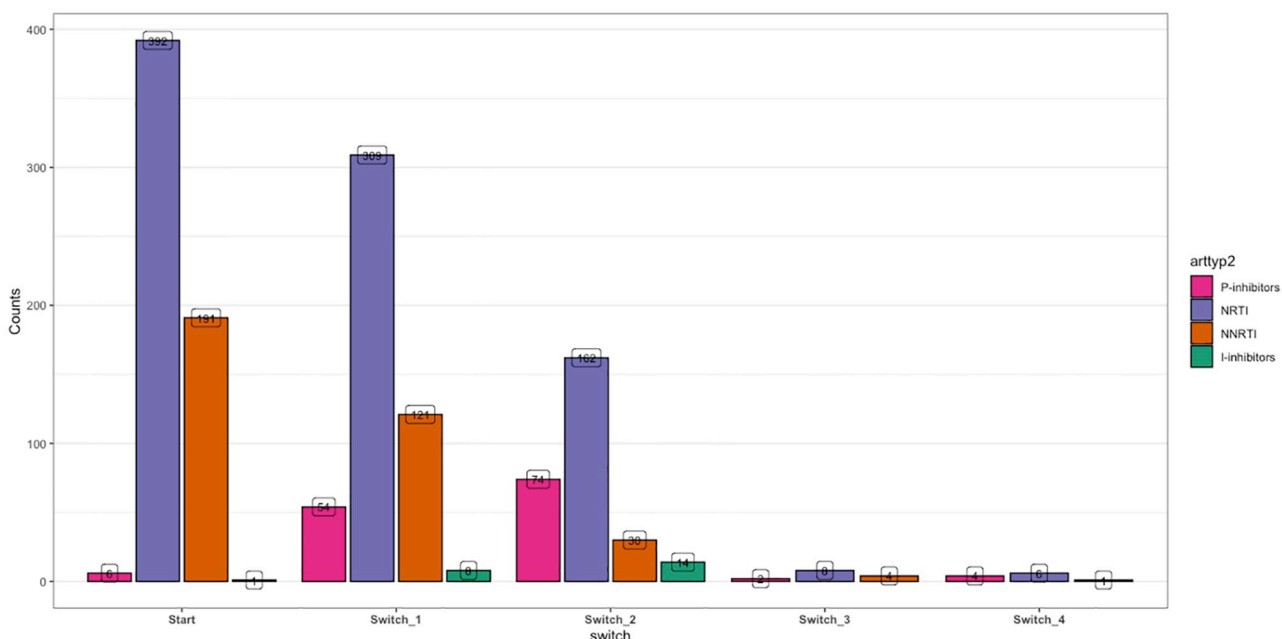

**Fig 2. ART treatment at each treatment switch.** Nucleoside reverse transcriptase inhibitors (NRTI), Non-nucleoside reverse transcriptase inhibitors (NNRTI), protease inhibitors (PI), and Integrase inhibitors (II).

it is important to note that the interaction between being female and having ever smoked was associated with significantly lower odds for observed low BM affecting the lumbar spine (OR<0.01, 95% CI <0.01 to 0.04).

## Discussion

In this cross-sectional study we observed that the duration of exposure to ART medication and ART classes have no significant effect on participants' risk for developing low BM and that high BMI is associated with reduced risk of developing low BMD in PLWH on long-term ART. People living with HIV are at increased risk of developing low BM when compared to age-matched HIV uninfected controls [34,35]. In this study, there were more cases of low BM as defined by the lumbar spine than the hip by a factor of 9. The prevalence of low BM was 18.1%. The prevalence of low BM in our study is consistent with what is reported in literature [8,9,17,19]. It is important to note that most longitudinal studies reporting on bone health in PLWH on long term ART use the WHO criteria for categorizing BMD instead of the ISCD official position that recommend the use of BM instead of BMD; despite having relatively young populations.

The ISCD also asserts that osteoporosis cannot be diagnosed in men under 50 years of age using BMD alone [20]. Therefore in the absence of local national guidance on the diagnosis and management of osteoporosis, we considered the official position of the International Society of Clinical Densitometry (ISCD) to categorize participants as having low BM or normal bone, since most participants were below 50 years of age [20]. However, the WHO criteria were also used to categorize post-menopausal females and males above 50 years. For ease of analysis, those categorized as osteopenia or osteoporosis were grouped together and considered as having low BM. The official position of International osteoporosis foundation (IOF) recommends BMD measurements at both the lumbar spine and the hip, and to use the lowest

**Table 3. Univariable and multivariable bone mass models.**

| | Univariate models | | Multivariate Model | | |
|---|---|---|---|---|---|
| | **Mean (SD)** | **95% CI** | **Mean (SD)** | **95% CI** | **N-Eff** |
| Dependent: Hip BM | | | | | |
| Duration on drug | 0.52 (1.96) | -3.34 to 4.52 | 0.77 (2.09) | -3.39 to 4.88 | 36670 |
| ART drug prescriptions | 0.00 (0.08) | -0.16 to 0.16 | 0.00 (0.08) | -0.16 to 0.17 | 52303 |
| Switch | -0.18 (0.50) | -0.78 to 1.20 | 0.10 (0.56) | -1.00 to 1.22 | 40584 |
| Age | -4.39 (4.79) | -14.24 to 4.79 | -3.56 (5.72) | -15.01 to 7.43 | 26296 |
| Gender as female | 0.14 (1.45) | -2.70 to 2.99 | 0.52 (5.27) | -9.89 to 10.84 | 25320 |
| Latest CD4 cell count | -2.75 (4.12) | -11.27 to 5.08 | -10.19 (15.30) | -40.78 to 19.82 | 24117 |
| Alcohol | 3.08 (2.66) | -1.34 to 9.04 | 3.00 (3.20) | -2.43 to 10.21 | 24362 |
| Ever smoked (yes) | -3.63 (3.64) | -11.92 to 2.20 | 3.75 (16.50) | -31.10 to 35.16 | 23220 |
| Virus suppressed (yes) | 3.56 (3.72) | -2.37 to 12.09 | 8.24 (16.04) | -21.85 to 41.85 | 23331 |
| Body mass index (BMI) | -2.80 (1.10) | -5.07 to -0.74* | -2.59 (6.30) | -15.01 to 7.43 | 23168 |
| *Under weight* | 4.25 (1.87) | 0.56 to 8.03* | - | - | - |
| *Normal weight (reference)* | 1 | 1 | - | - | - |
| *Overweight* | -2.24 (2.17) | -6.91 to 1.57 | - | - | - |
| *Obese* | -3.65 (4.34) | -14.00 to 3.12 | - | - | - |
| *Interactions* | | | | | |
| Age and ever smoked (yes) | - | - | -2.93 (26.02) | -59.14 to 42.59 | 22631 |
| Ever smoked and female gender | - | - | -1.60 (9.32) | -20.53 to 16.48 | 23201 |
| Ever smoked and Alcohol | - | - | -5.15 (8.41) | -22.61 to 11.08 | 24341 |
| Gender as female and BMI | - | - | 0.05 (2.48) | -4.74 to 4.97 | 23961 |
| Yes virus suppressed and BMI | - | - | -2.02 (5.93) | -13.46 to 10.10 | 23435 |
| CD4 and BMI | - | - | 4.16 (6.88) | -9.66 to 17.38 | 23341 |
| Dependent: Lumbar BM | | | | | |
| Duration on drug | -0.58 (1.84) | -3.04 to 4.11 | 0.72 (1.95) | -3.14 to 4.60 | 31708 |
| ART drug | 0.01 (0.06) | -0.13 to 0.12 | -0.00 (0.07) | -0.14 to 0.14 | 41214 |
| Switch | -0.10 (0.51) | -1.12 to 0.89 | -0.14 (0.56) | -1.26 to 0.95 | 36627 |
| Age | -15.26 (6.77) | -29.16 to -2.43* | -14.07 (7.69) | -29.43 to 0.84 | 13522 |
| Gender as female | -5.65 (2.05) | -9.82 to -1.76* | -4.01 (7.47) | -19.09 to 10.47 | 12560 |
| Latest CD4 cell count | -7.32 (5.88) | -19.10 to 4.06 | -19.93 (23.92) | -71.26 to 22.80 | 16327 |
| Alcohol | 3.03 (2.79) | -2.38 to 8.68 | 2.67 (3.36) | -3.85 to 9.45 | 14144 |
| Ever smoked (yes) | 3.56 (3.15) | -2.65 to 9.66 | 22.71 (16.59) | -9.30 to 55.92 | 14947 |
| Virus suppressed (yes) | 1.57 (3.52) | -5.15 to 8.56 | -3.57 (16.02) | -35.37 to 27.28 | 16868 |
| Body mass index | -5.47 (1.43) | -8.45 to -2.71* | -4.47 (6.15) | -16.94 to 7.18 | 15852 |
| *Under weight* | 9.56 (3.46) | 3.03 to 16.72* | - | - | - |
| *Normal weight (reference)* | 1 | 1 | - | - | - |
| *Overweight* | -6.07 (2.49) | -11.05 to -1.25* | - | - | - |
| *Obese* | -5.61 (4.09) | -14.03 to 1.92 | - | - | - |
| *Interactions* | | | | | |
| Age and ever smoked | - | - | -42.97 (32.41) | -110.02 to 15.76 | 16057 |
| CD4 cell & virus suppressed | - | - | 19.93 (24.78) | -24.88 to 72.83 | 16712 |
| Alcohol and ever smoked | - | - | -0.96 (7.26) | -15.12 to 13.51 | 13729 |
| Ever smoked and female yes | - | - | -22.28 (10.24) | -43.09 to -3.16* | 15585 |
| Viral suppression and BMI | - | - | -0.26 (5.96) | -11.56 to 11.94 | 16578 |
| BMI and Gender as female | - | - | 0.15 (3.16) | -5.95 to 6.45 | 13065 |

* P< 0.05.

BMD recorded [36–39]. The IOF official position further recommends that lumbar spine BMD be used for monitoring response to treatment while hip BMD is considered the best predictor for hip fractures [40].

The prevalence of low BM in our study is consistent with what was previously reported in a study conducted at our institution to investigate low BMD among patients failing first-line ART, though the mean duration on antiretroviral medication at the time was only 3.7 years [17]. With a mean ART experience of more than 12 years, we believe that our participants had stable BM. Antiretroviral therapy has been shown to significantly contribute to low BM irrespective of classes [9,10,14,41–43]; although this accelerated bone loss has been mainly associated with tenofovir disoproxil fumarate (TDF) containing regimens compared to non-TDF-containing regimens [14,43–45]. This initial bone loss is transient, possibly due to increased bone catabolism after viral load suppression and immune reconstitution [46–49], and has been shown to be most marked in the first two years of ART initiation with about 2–6% loss in BMD [50] followed by a long period of stable BMD or increase [51,52]. These changes in BMD over time have been shown to be comparable to those in uninfected health controls [51]. The pathogenesis of increased bone turn over in HIV infected individuals is rather complex and multifactorial [7]. HIV infection per se increases inflammatory cytokines that are thought to increase bone turn-over through the stimulation of osteoclast and bone resorption [7,53]. The mechanism of action of anti-retroviral drugs on low BM remains controversial [3,12]. Evidence from longitudinal data shows that among the different antiretroviral drugs, tenofovir is associated with bone loss [45,54–58] and this has been attributed to increased bone turnover. The exact mechanism by which TDF reduces bone mass is yet to be elucidated but, some studies have linked it to proximal renal tubular dysfunction [59–62] with possible resultant impairment in Vitamin D metabolism [59,61,63,64]. Protease inhibitors have been reported to suppress osteoclast function [65]. Several cross-sectional studies from LMIC have shown an association between non-nucleoside reverse transcriptase inhibitors (NNRTIs) particularly Efavirenz, which is widely used to treat HIV in LMIC, and low 25-hydroxyvitamin D [66–69]. Low levels of Vitamin D have been associated with a higher risk of HIV disease progression, death and virologic failure after ART [70–72]. Sub-optimal Vitamin D levels and cumulative exposure of tenofovir and protease inhibitors like lopinavir/ritonavir (LVP/RTV) have also been reported to independently increase the risk of low BMD [72,73] and fragility fractures [74]. TDF has been associated with a bigger decline in BMD than stavudine [55] or abacavir [45].

In the setting of long term viral suppression, understanding the complications of HIV infection and ART is paramount to successful management of HIV and its associated complications like low Bone mineral density [9]. Our findings further suggest that the duration of exposure to the various ART classes, is not associated with low BM; which is consistent with other studies which have reported stable BMD in PLWH on long-term ART [44,51,52,75–78], suggesting a long-term benefit of antiretroviral therapy. Bolland et. al [52] reported stable BMD in a cohort of 44 HIV infected middle-aged men on ART who were followed up for up to12 years. They further reported a BMD increase of 6.9% at the lumbar spine; and actually argued that it was not necessary to follow up men established on ART who do not have risk factors for BMD loss. Important to note, is the 5% mean increase in weight over the 12 years in their cohort. Increase in weight reduces the odds of developing low BMD, which is consistent with what was observed in our study. Bone loss is influenced by several factors including body weight, age, physical activity level, diet to mention a few [79]. Low BMI is a well-documented risk factor for osteoporosis and hip fractures in both men and women [10,80]. Weight loss is associated with 1% to 2% bone loss at the hip [81–83] and has been associated with bone mobilization and decrease in bone mineral content and BMD [84–86]. Therefore it is important for on ART to maintain adequate weight if they are to maintain good bone health.

In the setting of long term viral suppression, understanding the complications of HIV infection and ART is key to successful management of HIV and its associated complications [9]. We hope our results will raise awareness on bone health among clinicians and contribute to appropriate planning and management of low BMD among PLWH with the ultimate goal of making significant contribution towards reducing the NCD burden among this population.

## Strength of the study

The study was conducted among a well-characterized and organized cohort that had been consistently followed up every six months since 2004 and there was equal representation of both males and females. Secondly, all BMD scans were performed using the same machine and by the same technicians. Lastly, we utilized both Z-score and T-scores to diagnose low BMD in participants < 50 years and those above 50 years respectively as recommended by WHO [24,25].

## Limitations

This was a cross-sectional study conducted at one site; therefore, the study findings might not be generalizable to PLWH in the general population. Secondly, our results are based on only one DXA scan, participants had uniform exposure to ART and there was no control group; thus making it difficult to make any inferences on the effect of individual anti-retroviral drugs and regimens on bone loss.

## Conclusion

These results provide additional evidence that patients on long term ART achieve BM stabilization. It is important for PLWH on ART to maintain adequate weight if they are to maintain good bone health. There is need to develop local guidelines and protocols for the prevention and treatment of low bone mass not only in HIV, but also in the general public.

## Acknowledgments

We are very grateful to our research participants for their voluntary participation in this study.

## Author Contributions

**Conceptualization:** Erisa Sabakaki Mwaka, Ian Guyton Munabi, Andrew Kambugu, Elly Katabira.

**Data curation:** Erisa Sabakaki Mwaka, Ian Guyton Munabi, Barbara Castelnuovo, Arvind Kaimal, William Kasozi.

**Formal analysis:** Erisa Sabakaki Mwaka, Ian Guyton Munabi, Barbara Castelnuovo, Philippa Musoke, Elly Katabira.

**Funding acquisition:** Erisa Sabakaki Mwaka.

**Investigation:** Erisa Sabakaki Mwaka, Arvind Kaimal, William Kasozi, Philippa Musoke.

**Methodology:** Erisa Sabakaki Mwaka, Ian Guyton Munabi, Barbara Castelnuovo, Arvind Kaimal, William Kasozi, Philippa Musoke.

**Project administration:** Andrew Kambugu.

**Resources:** Andrew Kambugu.

**Supervision:** Barbara Castelnuovo, Andrew Kambugu, Philippa Musoke, Elly Katabira.

**Validation:** Erisa Sabakaki Mwaka.

**Writing – original draft:** Erisa Sabakaki Mwaka.

**Writing – review & editing:** Erisa Sabakaki Mwaka, Ian Guyton Munabi, Barbara Castel-nuovo, Arvind Kaimal, William Kasozi, Andrew Kambugu, Philippa Musoke, Elly Katabira.

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
