## [Decision Letter · Decision Letter 0]

18 Sep 2020

PONE-D-20-22805

Bone mineral density in people living with the HIV on long term antiretroviral therapy in Uganda

PLOS ONE

Dear Dr. Mwaka,

Thank you for submitting your manuscript to PLOS ONE. After careful consideration, we feel that it has merit but does not fully meet PLOS ONE’s publication criteria as it currently stands. Therefore, we invite you to submit a revised version of the manuscript that addresses the points raised during the review process.

ACADEMIC EDITOR: Please insert comments here and delete this placeholder text when finished. Be sure to:

Please respond to the three reviewers' comments and address them in the revised manuscript.The decision of acceptance or rejection is justified on PLOS ONE’s publication criteria.

We look forward to receiving your revised manuscript.

Kind regards,

Osama Farouk

Academic Editor

PLOS ONE

Journal Requirements:

2.We note that you have indicated that data from this study are available upon request. PLOS only allows data to be available upon request if there are legal or ethical restrictions on sharing data publicly. For information on unacceptable data access restrictions, please see http://journals.plos.org/plosone/s/data-availability#loc-unacceptable-data-access-restrictions.

3.Thank you for stating the following in the Financial Disclosure section:

[This research was supported by a sub-award to ESM (grant number 000503356-SP002-24) from the University of Alabama at Birmingham (UAB) Center For AIDS Research (https://www.uab.edu/medicine/cfar/) a National Institutes of Health funded program (P30AI027767-28, PI Michael Saags) that was made possible by NIAID/NIH/DHHS. The ALT cohort at the Infectious Disease Institute-Uganda is partly funded by Janssen pharmaceutical company through Johnson & Johnson corporate citizenship trust. The funders had no role in study design, data collection and analysis, decision to publish, or preparation of the manuscript.]. 

We note that you received funding from a commercial source: [Janssen pharmaceutical company]

Reviewers' comments:

Reviewer's Responses to Questions

**Comments to the Author**

1. Is the manuscript technically sound, and do the data support the conclusions?

Reviewer #1: Yes

Reviewer #2: Partly

Reviewer #3: Partly

2. Has the statistical analysis been performed appropriately and rigorously? 

Reviewer #1: Yes

Reviewer #2: I Don't Know

Reviewer #3: Yes

3. Have the authors made all data underlying the findings in their manuscript fully available?

Reviewer #1: Yes

Reviewer #2: No

Reviewer #3: Yes

4. Is the manuscript presented in an intelligible fashion and written in standard English?

Reviewer #1: Yes

Reviewer #2: Yes

Reviewer #3: Yes

5. Review Comments to the Author

Reviewer #1: The survey does give the general picture of BMD among a well-represented 200 PLWD. Since only one BMD was taken, the data did not provide information about the trend of development, particularly among the group with low BMD. To give real clinical value to the survey, another BMD needs to be arranged in 2-3 years' time to identify the changes so as to allow some practical recommendations on the management.

Reviewer #2: -how were the participants selected? What were the inclusion criteria? A consort flow diagram would be very helpful.

-Ref 22/23 are not appropriate: most participants were under 50 years old. Therefore, the ISCD (not the WHO!) recommends to apply the Z-scores, not the T-scores. Low bone mass is defined as a Z-scores <-2.0 meaning that long-term ART is not associated with low BMD. The categorisation of the measurements should be provided: how many participants had actually a Z-score below -2.0?

-discussion: the recommendations for screening are not supported by the data shown.

-conclusion: please do not over interprete our findings: association does not mean causality!

Reviewer #3: Thank you for your submission. I have the following comments:

1. Abstract: "This study set out to determine the prevalence and predictors of low BMD following long-term exposure to antiviral therapy....".

It seems to me that you have evaluated the prevalence of low BMD in your study but the predictors of low BMD were only studied in a very limited manner as there was no inclusion of lifestyle, exercise levels, education level, socio-economical and comorbidity, etc. The original statement needs revision to describe your completed work.

2. The sampling: need more description on random or convenient sampling from your pool of 8000 (refers page 5, line 8).

3. Table 1: need to include Alcohol data in your disclosure. For the BMI definition of overweight and obese vary between different countries, suggest to list out clearly.

4. Table 3: suggest to add a bar chart presentation for clarity.

5. Table 4: need to improve the presentation, plus add more explanation and justification.

6. Having revised 1 to 5, the title of the paper could be modified to reflect your actual completed work.

6. PLOS authors have the option to publish the peer review history of their article (what does this mean?). If published, this will include your full peer review and any attached files.

Reviewer #1: No

Reviewer #2: No

Reviewer #3: **Yes: **Anthony WL KWOK, PhD

---

## [Author Response · Author response to Decision Letter 0]

12 Nov 2020

Response to Editor

1. a) Comment: Please ensure that your manuscript meets PLOS ONE's style requirements, including those for file naming. 

Response: The manuscript has been formatted and meets the PLOS ONE style.

b) Comment: If there are ethical or legal restrictions on sharing a de-identified data set, please explain them in detail (e.g., data contain potentially identifying or sensitive patient information) and who has imposed them (e.g., an ethics committee). Please also provide contact information for a data access committee, ethics committee, or other institutional body to which data requests may be sent.

Response: There are no ethical or legal restrictions to sharing a de-identified set of our data. The dataset has been submitted and published on Figshare, DOI:10.6084/m9.figshare.13174628.

c) Comment: If there are no restrictions, please upload the minimal anonymized data set necessary to replicate your study findings as either Supporting Information files or to a stable, public repository and provide us with the relevant URLs, DOIs, or accession numbers.

Response: This has been done.

2. Comment: We note that you received funding from a commercial source: [Janssen pharmaceutical company]. Please provide an amended Competing Interests Statement that explicitly states this commercial funder, along with any other relevant declarations relating to employment, consultancy, patents, products in development, marketed products, etc.

Response: This research was supported by a sub-award to ESM (grant number 000503356-SP002-24) from the University of Alabama at Birmingham (UAB) Center For AIDS Research (https://www.uab.edu/medicine/cfar/) a National Institutes of Health funded program (P30AI027767-28, PI Michael Saags) that was made possible by NIAID/NIH/DHHS. The ALT cohort at the Infectious Disease Institute-Uganda is partly funded by Janssen pharmaceutical company through Johnson & Johnson corporate citizenship trust. The funders had no role in study design, data collection and analysis, decision to publish, or preparation of the manuscript. We declare competing interest; Janssen pharmaceutical company provides financial support to the Antiretroviral Treatment Longterm (ALT) Cohort at the Infectious Diseases Institute-Uganda. However, this does not alter our adherence to PLOS ONE policies on sharing data and materials.

Response to reviewers

Reviewer #1

Comment: The survey does give the general picture of BMD among a well- represented 200 PLWD. Since only one BMD was taken, the data did not provide information about the trend of development, particularly among the group with low BMD. To give real clinical value to the survey, another BMD needs to be arranged in 2-3 years' time to identify the changes so as to allow some practical recommendations on the management

Response: We are very interested in longitudinal follow up of the bone health of these individuals and are therefore working on a grant application to include bone health as one of the routine annual follow up procedures for the cohort. If successful we hope to follow them up for up to 96 weeks.

Reviewer #2

Comment: How were the participants selected? What were the inclusion criteria? A consort flow diagram would be very helpful.

Response: We have added a statement “Using computer generated numbers we randomly enrolled adults above 18 years who had been receiving ART for at least 10 years and were willing and able to comply with all study procedures” Lines 87-89. We have also added a flow diagram as suggested.

Comment: Ref 22/23 are not appropriate: most participants were under 50 years old. Therefore, the ISCD (not the WHO!) recommends to apply the Z-scores, not the T-scores. Low bone mass is defined as a Z-scores <-2.0 meaning that long-term ART is not associated with low BMD. The categorisation of the measurements should be provided: how many participants had actually a Z-score below -2.0?

Response: We have re-analysed the data using the official position of the International Society of Clinical Densitometry (ISCD) since most participants were below 50 years of age. A Z-score of -2.0 or lower has been defined as having low bone mass for age and a Z-score above -2.0 has been defined as bone mass within the expected range for age. The World Health Organization has been used for post-menopausal females and males above 50 years of age. For ease of data analysis, all participants with low bone mass or osteopenia/osteoporosis have been categorized as having “low bone mass” while the rest are categorized as having normal bone mass (BM). The term “Bone Mass” has been adopted in place of “Bone Mineral Density”. 4/199 and 36/199 have either Z-scores below -2.0 or T-score below -1.0 for the hip and lumbar spine respectively.

The title has also been changed to “Low bone mass in people living with HIV on long-term anti-retroviral therapy in Uganda”

Comment: discussion: the recommendations for screening are not supported by the data shown.

Response: The recommendation on screening has been deleted.

Comment: conclusion: please do not over interpret our findings: association does not mean causality!

Response: The conclusion has been recast.

Reviewer #3

Comment: Abstract: "This study set out to determine the prevalence and predictors of low BMD following long-term exposure to antiviral therapy....".

It seems to me that you have evaluated the prevalence of low BMD in your study but the predictors of low BMD were only studied in a very limited manner as there was no inclusion of lifestyle, exercise levels, education level, socio-economical and comorbidity, etc. The original statement needs revision to describe your completed work.

Response: This has been revised to “This study set out to determine the prevalence of low bone mass following long-term exposure to antiretroviral therapy in Ugandan people living with HIV”.

Comment: The sampling: need more description on random or convenient sampling from your pool of 8000.

Response: A better description is given and a flow diagram included (Fig 1).

Comment: Table 1: need to include Alcohol data in your disclosure. For the BMI definition of overweight and obese vary between different countries, suggest to list out clearly.

Response: A statement had been added with appropriate references “The WHO general cut off points for BMI classification were used for this study however literature suggests that they may vary with race/ethnicity and between countries”.

Comment: Table 3: suggest to add a bar chart presentation for clarity.

Response: Table 3 has been replaced by a bar chart (Fig 2).

Comment: Table 4: need to improve the presentation, plus add more explanation and justification

Response: More information has been provided in the narrative to ease interpretation of Table 4 (Now Table 3).

Comment: 6. Having revised 1 to 5, the title of the paper could be modified to reflect your actual completed work.

Response: The title has been revised to “Low bone mass in people living with HIV on long-term anti-retroviral therapy in Uganda”.

---

## [Decision Letter · Decision Letter 1]

24 Dec 2020

PONE-D-20-22805R1

Low bone mass in people living with the HIV on long term antiretroviral therapy in Uganda

PLOS ONE

Dear Dr. Mwaka,

Thank you for submitting your manuscript to PLOS ONE. After careful consideration, we feel that it has merit but does not fully meet PLOS ONE’s publication criteria as it currently stands. Therefore, we invite you to submit a revised version of the manuscript that addresses the points raised during the review process.

Please, revise the title of the article as recommended by the reviewers.

We look forward to receiving your revised manuscript.

Kind regards,

Osama Farouk

Academic Editor

PLOS ONE

Reviewers' comments:

Reviewer's Responses to Questions

**Comments to the Author**

1. If the authors have adequately addressed your comments raised in a previous round of review and you feel that this manuscript is now acceptable for publication, you may indicate that here to bypass the “Comments to the Author” section, enter your conflict of interest statement in the “Confidential to Editor” section, and submit your "Accept" recommendation.

Reviewer #1: All comments have been addressed

Reviewer #2: All comments have been addressed

Reviewer #3: All comments have been addressed

2. Is the manuscript technically sound, and do the data support the conclusions?

Reviewer #1: Yes

Reviewer #2: Yes

Reviewer #3: Yes

3. Has the statistical analysis been performed appropriately and rigorously? 

Reviewer #1: N/A

Reviewer #2: Yes

Reviewer #3: Yes

4. Have the authors made all data underlying the findings in their manuscript fully available?

Reviewer #1: Yes

Reviewer #2: Yes

Reviewer #3: Yes

5. Is the manuscript presented in an intelligible fashion and written in standard English?

Reviewer #1: Yes

Reviewer #2: Yes

Reviewer #3: Yes

6. Review Comments to the Author

Reviewer #1: (No Response)

Reviewer #2: (No Response)

Reviewer #3: The revised information have much improved the accuracy and quality of the first draft. The additional information in the tables and charts are relevant and clear that explains the original idea better.

As your study is a single centre study with all the data from the same centre, you can think about to fine tune the title further as below: Low bone mass in people living with the HIV on long term antiviral therapy: a single centre study in Uganda

7. PLOS authors have the option to publish the peer review history of their article (what does this mean?). If published, this will include your full peer review and any attached files.

Reviewer #1: **Yes: **Leung Ping Chung

Reviewer #2: No

Reviewer #3: **Yes: **Anthony WL Kwok

---

## [Author Response · Author response to Decision Letter 1]

30 Dec 2020

Comment: We note that you have included tables 1,2 and 4 but there is not Table 3, if this should be included then please include a copy within your manuscript and cite in your text. - or. If table 4 should actually be Table 3 then please amend the numbering of the table and citations within your text

Response: Thanks very much for this observation. Table 3 was mislabeled as Table 4. This has been rectified. “Table 4” now appears as “Table 3”.

Comment: Please amend the title either on the online submission form or in your manuscript so that they are identical.

Response: This has been done

---

## [Decision Letter · Decision Letter 2]

18 Jan 2021

Low bone mass in people living with HIV on long term antiretroviral therapy: a single centre study in Uganda

PONE-D-20-22805R2

Dear Dr. Mwaka,

We’re pleased to inform you that your manuscript has been judged scientifically suitable for publication and will be formally accepted for publication once it meets all outstanding technical requirements.

Kind regards,

Osama Farouk

Academic Editor

PLOS ONE

Additional Editor Comments (optional):

Reviewers' comments:

Reviewer's Responses to Questions

**Comments to the Author**

1. If the authors have adequately addressed your comments raised in a previous round of review and you feel that this manuscript is now acceptable for publication, you may indicate that here to bypass the “Comments to the Author” section, enter your conflict of interest statement in the “Confidential to Editor” section, and submit your "Accept" recommendation.

Reviewer #3: All comments have been addressed

2. Is the manuscript technically sound, and do the data support the conclusions?

Reviewer #3: Yes

3. Has the statistical analysis been performed appropriately and rigorously? 

Reviewer #3: Yes

4. Have the authors made all data underlying the findings in their manuscript fully available?

Reviewer #3: Yes

5. Is the manuscript presented in an intelligible fashion and written in standard English?

Reviewer #3: Yes

6. Review Comments to the Author

Reviewer #3: In your revised copy, you addressed all the issues that I have raised earlier. I have no additional comment to add. The title of the paper looks appropriate and accurate now.

7. PLOS authors have the option to publish the peer review history of their article (what does this mean?). If published, this will include your full peer review and any attached files.

Reviewer #3: **Yes: **Anthony WL Kwok

---

## [Editor Report · Acceptance letter]

28 Jan 2021

PONE-D-20-22805R2 

Low bone mass in people living with HIV on long-term anti-retroviral therapy: a single centre study in Uganda 

Dear Dr. Mwaka:

I'm pleased to inform you that your manuscript has been deemed suitable for publication in PLOS ONE. Congratulations! Your manuscript is now with our production department. 

Kind regards, 

on behalf of

Dr. Osama Farouk 

Academic Editor

PLOS ONE